# SPG-SAM: Semantic Prompt Graph Learning for Multi-Class Medical Image Segmentation

## Abstract

Existing visual foundation model-based methods (e.g., SAM) for multi-class medical image segmentation typically face a trade-off between insufficient semantic information and spatial prompt interference, while extending SAM with fully automated semantic segmentation compromises its inherent interactive prompting capabilities. To bridge the semantic specificity gap, we propose SPG-SAM (Semantic Prompt Graph learning for SAM), a novel framework that seamlessly integrates spatial and semantic prompting for efficient and accurate multi-class medical image segmentation. SPG-SAM introduces dedicated semantic prompts to complement SAM's spatial prompts, establishing an explicit mapping between object locations and semantic categories. Furthermore, we introduce a semantic prompt graph learning module that employs a graph attention network to explicitly model anatomical priors and structural relationships among medical objects. This design enables cross-category feature interaction, mitigates prompt interference, and facilitates accurate and efficient multi-class segmentation within the SAM-based paradigm. Experimental results demonstrate that SPG-SAM achieves average Dice coefficients of 94.27% and 91.83% on the abdominal multi-organ segmentation (BTCV) and pelvic target segmentation (PelvicRT) tasks, respectively, outperforming the second-best state-of-the-art baselines by 2.1% and 3.65%. Code available at XXX.

## 1 Introduction

Medical image segmentation classifies pixels or voxels in modalities like CT, MRI, and ultrasound into predefined anatomical or lesion regions (e.g., organs, tumors), serving as a fundamental medical image analysis task (Litjens et al., 2017; Wang et al., 2022; Norouzi et al., 2014; Azad et al., 2024; Siddique et al., 2021). The Segment Anything Model (SAM) (Kirillov et al., 2023) introduced prompt learning in open-set visual segmentation, establishing a "visual foundation model + interactive prompting" framework that sparked a new era in foundation model-based medical image segmentation. However, SAM and similar models, trained mainly on natural images, exhibit significant performance gaps in medical imaging (Ravi et al., 2024; Chen et al., 2023; Wu et al., 2025; Lin et al., 2024). They lack intrinsic understanding of medical imaging physics and anatomical constraints, often misinterpreting complex tissues, and their single-target segmentation struggles with multi-target medical scenarios, such as multi-organ localization in abdominal or pelvic CT scans (Du et al., 2020; Liu et al., 2021; Yuan et al., 2023).

To address these, approaches like MedSAM (Ma et al., 2024) and SAMed (Zhang & Liu, 2023) adapt SAM via domain-specific pretraining or mask decoder modifications for semantic segmentation. Yet, they treat multi-class segmentation and interactive prompting as conflicting goals: SAM's prompts support only single-target annotation without semantics, while multi-class requires explicit semantic constraints. Maintaining SAM's interaction demands multiple prompts for multi-organ tasks, reducing clinical efficiency; fully automated semantic segmentation loses fine-grained user control vital in medicine (Zhang & Liu, 2023; Li et al., 2023).

We propose SPG-SAM (Semantic Prompt Graph learning for SAM), a framework integrating spatial and semantic prompts to enable efficient, accurate multi-class medical segmentation while preserving interactivity. Semantic prompts complement SAM's spatial prompts by explicitly mapping locations to semantic categories, allowing the model to identify both *where* and *what* in dense, multi-structure scenarios. This dual prompting mitigates interference and leverages SAM's design strengths.

Further, SPG employs a Graph Attention Network (GAT) (Velickovic et al., 2017) to model inter-class anatomical relations as nodes and weighted edges, using semantic prompts as contextual priors. Multi-layer attention learns semantic correlations among adjacent anatomical structures, enabling cross-category knowledge transfer and enhancing segmentation robustness in ambiguous or overlapping regions.

The main contributions of our work are summarized as follows (the key architectural difference between our method and prior methods is also highlighted in Figure 6 in the Appendix):

- We propose SPG-SAM, a novel SAM-based medical image segmentation framework that introduces semantic prompts alongside spatial prompts to enable accurate and efficient multi-class segmentation while preserving SAM's interactive control.
- We further enhance semantic understanding through semantic prompt graph learning, which leverages a graph attention network to model inter-class relationships among medical targets, capturing anatomically informed priors and facilitating cross-category feature interaction.
- Our approach achieves state-of-the-art performance on multiple multi-class medical image segmentation benchmarks, including the BTCV (Landman et al., 2015) and PelvicRT datasets, delivering 2.1% and 3.65% improvements in Dice score for segmentation accuracy, respectively.

## 2 RELATED WORK

### 2.1 MEDICAL IMAGE SEGMENTATION METHODS

Early studies used CNNs (LeCun et al., 1998), with U-Net (Ronneberger et al., 2015) excelling in localization via skip connections. Later models like TransUNet (Chen et al., 2021) and SwinUNet (Cao et al., 2022) added Transformer attention to improve segmentation of irregular organs. Yet, they still struggle with semantic reasoning and generalization in multi-class tasks (Zhou et al., 2019; Xiao et al., 2018; Guan et al., 2019). Recent advances in foundation models, particularly the SAM, have revolutionized visual segmentation by introducing a promptable architecture that flexibly adapts to various open-set segmentation tasks, significantly enhancing the model's applicability across different scenarios. In the field of medical image segmentation, to bridge the inherent gap between medical and natural images, existing adaptation methods primarily fall into two categories: fine-tuning SAM on large-scale medical datasets while retaining the single-class segmentation paradigm (Azad et al., 2023; Cheng et al., 2023), and modifying SAM's decoder to enable multi-class prediction (Zhang & Liu, 2023; Li et al., 2023). Although these approaches improve SAM's applicability in medical imaging, they either sacrifice interactivity or fail to model cross-category dependencies, leading to prompt interference issues in multi-organ segmentation.

### 2.2 GRAPH ATTENTION NETWORKS

Graph Attention Networks (GATs) (Velickovic et al., 2017) overcome the limitations of traditional Euclidean data modeling by introducing self-attention mechanisms on graph-structured data. Unlike graph convolutional networks (GCNs) (Kipf & Welling, 2016), which use fixed-weight neighborhood aggregation, GATs dynamically compute attention coefficients between nodes, enabling adaptive feature propagation based on node importance. This architecture has been applied in medical image analysis tasks to capture structured information from anatomical priors or multi-scale features (Foo et al., 2022; Zhang, 2023). However, existing methods often treat inter-class relationships as static or task-specific attributes, limiting their generalization across diverse anatomical structures.

## 3 METHOD

### 3.1 OVERVIEW

To address the semantic underspecification issue in SAM while preserving its interactive prompting capability, we introduce SPG-SAM, which supplements categorical information as semantic prompts to bridge the semantic specificity gap. Furthermore, we leverage these semantic prompts through a graph attention network, explicitly modeling anatomical priors and structural relationships between medical objects. The main structure of our proposed SPG-SAM is illustrated in Figure 1.

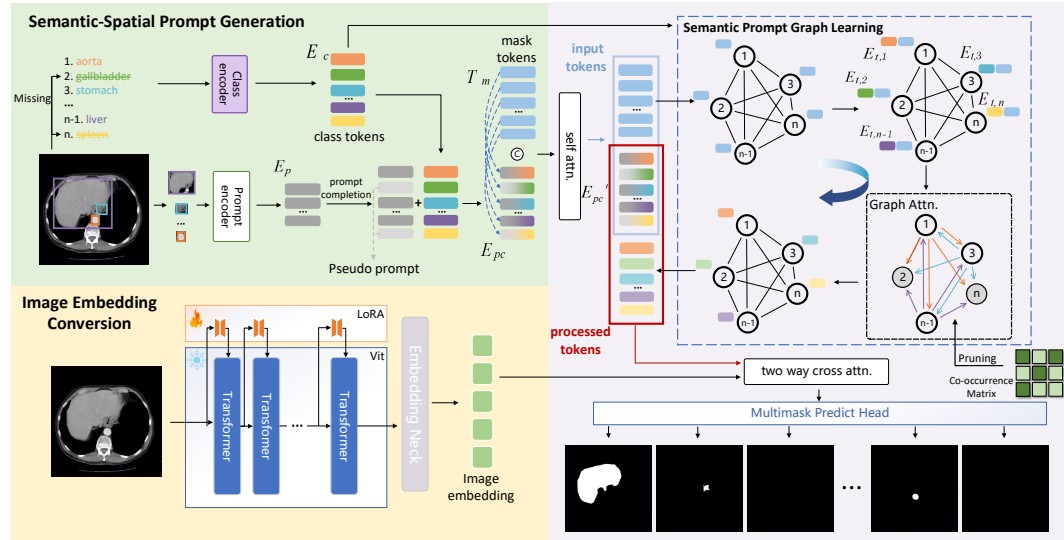

Figure 1: The SPG-SAM framework comprises three stages. **Stage 1**: Prompt Encoding and Visual Feature Extraction. Spatial prompts $E_p^i$ and category embeddings $E_c^i$ are coupled via a joint encoder to form semantic-spatial embeddings $E_{pc}$ ($E_p^i + E_c^i \rightarrow E_{pc}$), while the image encoder extracts visual features $I$. **Stage 2**: Semantic Prompt Graph Learning and Anatomical Constraint Modeling. The GAT-based SPG module interacts $E_{pc}$ with visual features to dynamically construct an anatomical adjacency matrix $A_{ij}$. Cross-category relationships are modeled through node collaboration $T_i''$, with a category pruning strategy ($A_{ij} \in \{0, 1\}$) suppressing interference from missing categories. **Stage 3**: Multi-target Collaborative Decoding. Features $T_i''$ are mapped via $\text{MLP}_i$ to $T_i'''$, then matrix-multiplied with $I^\top$ to generate multi-channel logits, and finally activated by Sigmoid to output anatomically constrained multi-category segmentation masks $M_{\text{pred}}$.

## 3.2 Augmenting Spatial Prompts with Semantic Prompts

The original spatial prompts (e.g., points, bounding boxes) in SAM lack semantic information, leading to ambiguous localization, mutual interference among multi-class instances, and erroneous activation in multi-class segmentation scenarios. SPG-SAM addresses this through a coordinated encoding scheme that integrates spatial prompts in native coordinate systems with semantic prompts. This framework employs pseudo-prompt filling for missing semantics to maintain consistent mapping relationships, while implicitly establishing topological correlations between geometric cues and semantic categories to enhance segmentation specificity.

**Pseudo-Prompt Filling for Missing Classes** In a given medical segmentation task, the target categories are typically predefined and remain fixed. However, not all categories are consistently present in every image (e.g., each 2D slice within a 3D volume), leading to missing prompts for absent classes. To ensure that each category receives appropriate guidance prompts corresponding to fixed output channels, SPG-SAM employs pseudo-prompts to compensate for missing entries. Specifically, when a particular class is absent, we supplement it with a fixed pseudo-prompt (i.e., $p_{-1}$) as its category-specific prompt for that image, which is then fed into the prompt encoder along with other prompts:

$$E_p^i = Enc_{\text{prompt}}(p_i), \quad p_i = \begin{cases} p_i & i \in \mathcal{C}' \\ p_{-1} & i \notin \mathcal{C}' \end{cases} \tag{1}$$

where $E_p$ is the spatial prompt embeddings, $\mathcal{C}'$ denotes present categories, $p_i$ represents class-specific sparse prompts, and $Enc_{\text{prompt}}(\cdot)$ is the original prompt encoder. This guarantees a stable spatial-semantic mapping.

**Semantics Encoding** Medical anatomical categories exhibit explicit definitions and fixed cardinality, demonstrating strong target-specific guidance. To incorporate target-specific semantic information

as supplementary guidance for existing spatial prompts, we introduce a category encoder $Enc_{\text{cat}}(\cdot)$ that transforms categorical information into semantic embeddings:

$$E_c^i = Enc_{\text{cat}}(\mathcal{C}_i; \theta_{cat}), \quad \forall i \in \{1, \ldots, N\} \tag{2}$$

where $N$ represents the total number of target classes, and $Enc_{\text{cat}}(\cdot)$ is the category encoder. We construct semantic prompt embeddings $E_c$ as supplementary semantic-layer guidance to spatial prompts. The constructed $E_c$ possesses three essential characteristics: 1) Cardinality equivalence to the medical structures in the target dataset, ensuring one-to-one correspondence; 2) All $E_c$ maintain a unified embedding dimension $d_c$, compatible with the embedding space of spatial prompts; 3) Learnable parameters optimized through backpropagation to adaptively meet segmentation requirements.

**Semantic Prompt Construction**    To guide the prompts for segmenting specific organs or lesions, we design a prompt coupling strategy inspired by SAM's classification mechanism for prompt categories. The spatial prompt embedding $E_p$ is integrated with its corresponding category embedding to generate the spatial-category embedding $E_{pc}$:

$$E_{pc} = \mathop{\Big|\Big|}_{i=1}^{N} \left( E_c^i + E_p^i \right). \tag{3}$$

The processed and concatenated spatial-category embedding incorporates explicit target category information, thereby distinguishing it from other prompts. Through this approach, we introduce "category" as a semantic prompt type for SPG-SAM while coupling it with spatial prompts. This strategy enhances the correspondence between prompts and their target segmentation categories, enabling the spatial-category embedding to replace raw prompts in subsequent decoding steps.

## 3.3    SEMANTIC PROMPT GRAPH LEARNING

**Graph Structure**    Medical images from the same series often exhibit similar anatomical relationships, such as fixed categories of medical objects, stable relative positions of objects, and regular co-occurrence patterns. The entire 2D medical image can be abstracted as a fully connected undirected graph $G = (V, E)$, where the vertices $V$ represent the various medical objects to be segmented, and the potential relationships between objects form the edge set $E$. The number of vertices in the graph $G$ remains constant, and the meaning of each vertex is fixed, while the edge weights can be dynamically adjusted based on the relationships between adjacent objects. In our framework, each vertex in the graph (representing a type of medical object) is connected to its neighboring vertices via weighted edges. This connection effectively expresses their interdependence and spatial relationships.

**Prompt Graph Learning**    To enhance the utilization of global structural information from the images, SPG-SAM introduces a semantic prompt graph learning module to learn the implicit relationship graph $G_{ir} = (V_{ir}, E_{ir})$ between various medical objects. Here, $V_{ir}$ is the set of all mask tokens $T$ for medical objects to be segmented, and $E_{ir}$ does not need to be predefined but is implicitly generated through weight self-learning during network training.

During the construction of the graph $G_{ir}$, we inject corresponding semantic information into the graph attention mechanism (Velickovic et al., 2017; Li et al., 2019) to establish a deterministic mapping relationship. By providing targeted category information as guidance, the originally directionally ambiguous mask tokens acquire clear categorical meanings. Specifically, we concatenate the learnable mask token $T_i \in \mathbb{R}^{d_t}$ with the corresponding semantic prompt embeddings $E_c$ along the last channel, represented as:

$$T_i' = [T_i || E_c^i], \quad i \in \{1, \ldots, N\} \tag{4}$$

where $T_i' \in \mathbb{R}^{d_t + d_c}$, and $d_t$ and $d_c$ denote the vector lengths of $T_i$ and $E_c^i$.

Subsequently, a shared self-attention mechanism is executed for each node, dynamically adjusting the node's features based on its adjacent nodes' features, integrating global features into individual node features. The self-attention mechanism for each node is:

$$T_i'' = \mathop{\Big|\Big|}_{k=1}^{K} \sigma \left( \sum_{j \in \mathcal{N}_i} \alpha_{ij}^k W^k T_j' \right). \tag{5}$$

where $\sigma(\cdot)$ is a non-linear activation function, $W \in \mathbb{R}^{d_t \times (d_t + d_c)}$ is a mapping matrix ensuring the output channel is consistent with the input mask token, and $K$ is for the multi-head attention mechanism. $\alpha_{ij}$ is the attention coefficient between nodes, expressed as:

$$\alpha_{ij} = softmax_j(e_{ij}) = \frac{exp(e_{ij})}{\sum_{n \in \mathcal{N}_i} exp(e_{in})} \tag{6}$$

where $\mathcal{N}_i$ is the set of neighboring nodes of node $i$, and $e_{ij}$ represents the correlation between different objects using the scaled dot product: $e_{ij} = (W_q W T_i')^T \cdot (W_k W T_j')$, where $W_q$ and $W_k \in \mathbb{R}^{d_t \times d_t}$ are query mapping matrix and key mapping matrix.

To capture richer semantic information, SPG-SAM adopts a multi-head attention mechanism, parallelizing $K$ independent attention mechanisms and concatenating the output features to form attention: The generated attention $T_i''$ is residually connected with the original mask token $T_i$ to obtain an enhanced mask token.

**Category Pruning** Since medical objects do not always appear in every image to be segmented, and interactions should only occur between objects within the current image, SPG-SAM incorporates a category pruning strategy within the graph attention mechanism to effectively eliminate the influence of absent objects on present ones. Specifically, we construct a co-occurrence-based adjacency matrix $\mathbf{A}$ to represent whether any two objects appear simultaneously.

When constructing the adjacency matrix, each element $\mathbf{A}_{ij}$ indicates whether the two medical objects $i$ and $j$ co-occur in the same image. If both objects appear together, then $\mathbf{A}_{ij} = 1$; otherwise, $\mathbf{A}_{ij} = 0$. In this context, the attention generation process in Eq. (5) is updated to:

$$T_i'' = \bigg\|_{k=1}^{K} \sigma \left( \sum_{j \in \mathcal{N}_i} \mathbf{A}_{ij} \alpha_{ij}^k W^k T_j' \right). \tag{7}$$

This co-occurrence-based adjacency matrix not only provides the model with a clear category relationship graph, optimizing the flow of information within the graph attention mechanism to ensure that only relevant objects influence each other, thereby improving the accuracy and robustness of the segmentation, but also reduces redundant computations through pruning, enhancing the model's focus on the categories that are actually present.

### 3.4 MULTI-CLASS PREDICTION MASK OUTPUT

In terms of architecture, we adopt a prediction head construction method similar to SAMed (Zhang & Liu, 2023) to simultaneously output segmentation masks for different categories. Specifically, after the fusion process, each updated mask token embedding is adjusted in channel size through a 3-layer MLP:

$$T_i''' = \text{MLP}_i(T_i''), \quad i \in \{1, 2, \ldots, N\}. \tag{8}$$

where the term $\text{MLP}_i$ represents the MLP corresponding to each mask token. Subsequently, $T_i'''$ is spatially dot-multiplied with the image embedding to obtain the predicted segmentation logits for the corresponding class:

$$M_{logits,i} = T_i''' \cdot I^\top. \tag{9}$$

Finally, the predicted segmentation logits for all classes are concatenated and passed through a sigmoid function to produce the final segmentation result:

$$M_{pred} = \text{Sigmoid}(M_{logits}), \tag{10}$$

where $M_{logits} = [M_{logits,1}, M_{logits,2}, \ldots, M_{logits,n}]$. We employ the same loss function as in SAM for training, which is formulated as follows:

$$L = \alpha L_{BCE}(M_{logits}, M_{gt}) + (1 - \alpha) L_{Dice}(M_{logits}, M_{gt}). \tag{11}$$

Here, $M_{gt}$ represents the ground truth corresponding to $M_{logits}$, while $L_{BCE}$ and $L_{Dice}$ denote the binary cross-entropy loss and Dice loss, respectively, which are balanced with the hyperparameter $\alpha$.

Table 1: Performance on the BTCV dataset. SAM, MedSAM, and SAM-Med2D retain interactive prompt functionality, with the prompt format being the ground truth bounding box of the target segmentation region. Bold and underlined numbers indicate the best and second-best scores. Organ abbreviations: Gallb. (Gallbladder), K(L) (Left Kidney), K(R) (Right Kidney), Pancr. (Pancreas), Stom. (Stomach).

| Methods | AVE DSC (%) | Aorta | Gallb. | K(L) | K(R) | Liver | Pancr. | Spleen | Stom. |
|---|---|---|---|---|---|---|---|---|---|
| Unet (Ronneberger et al., 2015) | 84.66 | 82.24 | 86.20 | 88.44 | 91.84 | 87.52 | 76.68 | 86.21 | 78.12 |
| TransUnet (Chen et al., 2021) | 82.46 | 86.22 | 87.42 | 87.91 | 86.62 | 88.16 | 56.12 | 87.48 | 79.76 |
| SwinUnet (Cao et al., 2022) | 84.16 | 84.30 | 88.67 | 92.58 | 89.01 | 89.81 | 57.03 | 90.24 | 81.66 |
| MissFormer (Huang et al., 2021) | 82.97 | 79.15 | 88.76 | 89.79 | 87.28 | 87.81 | 64.89 | 84.84 | 81.21 |
| TransDeepLab (Azad et al., 2022) | 84.29 | 85.95 | 86.86 | 90.89 | 88.31 | 88.73 | 60.75 | 89.84 | 83.01 |
| HiFormer (Heidari et al., 2023) | 84.08 | 86.61 | 90.01 | 86.85 | 86.85 | 90.34 | 58.91 | 89.58 | 83.48 |
| DAEFormer (Azad et al., 2023) | 84.41 | 83.34 | 88.66 | 89.30 | 89.54 | 90.77 | 67.37 | 89.45 | 76.87 |
| SAM (Kirillov et al., 2023) | 87.86 | 90.06 | 92.45 | 88.56 | 87.76 | 89.25 | 85.47 | 90.12 | 79.20 |
| nnSAM (Li et al., 2023) | 87.62 | 88.16 | 92.60 | 86.13 | 87.72 | 91.85 | 86.34 | 92.61 | 75.53 |
| MedSAM (Ma et al., 2024) | 90.44 | _91.17_ | 91.67 | 91.95 | 91.95 | 93.70 | _87.36_ | 91.17 | 84.55 |
| SAM-Med2D (Cheng et al., 2023) | _92.17_ | 88.74 | _94.14_ | _94.69_ | _93.60_ | _94.10_ | 86.18 | _95.96_ | _89.95_ |
| SAM2UNet (Xiong et al., 2024) | 86.70 | 83.37 | 90.63 | 86.40 | 89.56 | 90.12 | 81.00 | 92.19 | 80.36 |
| SPG-SAM (Ours) | **94.27** | **93.30** | **96.42** | **95.05** | **94.34** | **94.60** | **92.55** | **96.50** | **91.36** |

Table 2: Performance on the PelvicRT dataset. The details are the same as in Table 1. Organ abbreviations: CTV (Clinical Target Volume), FH(L) (Left Femoral Head), FH(R) (Right Femoral Head), SI (Small Intestine).

| Methods | AVE DSC (%) | Bladder | Colon | CTV | FH(L) | FH(R) | Rectum | SI |
|---|---|---|---|---|---|---|---|---|
| Unet (Ronneberger et al., 2015) | 82.14 | 92.73 | 75.07 | 80.28 | 92.49 | 90.39 | 71.02 | 73.01 |
| TransUnet (Chen et al., 2021) | 87.17 | **96.02** | 72.58 | 83.69 | 95.03 | 93.67 | 91.57 | 77.61 |
| SwinUnet (Cao et al., 2022) | 84.65 | 93.72 | 68.88 | 79.25 | 93.14 | 94.11 | 85.68 | 77.75 |
| MissFormer (Huang et al., 2021) | 77.52 | 95.59 | 64.19 | 74.31 | 80.12 | 81.06 | 80.27 | 67.08 |
| TransDeepLab (Azad et al., 2022) | 83.81 | 90.00 | 71.23 | 75.79 | 94.35 | 92.24 | 90.14 | 72.90 |
| HiFormer (Heidari et al., 2023) | 86.90 | 95.42 | 73.12 | 80.54 | 95.72 | 96.03 | 91.20 | 76.24 |
| DAEFormer (Azad et al., 2023) | 81.72 | 91.12 | 68.10 | 77.15 | 82.71 | 85.48 | 90.74 | 76.77 |
| SAM (Kirillov et al., 2023) | 87.90 | 90.41 | 73.38 | 85.45 | **97.24** | 92.40 | _92.29_ | 84.16 |
| nnSAM (Li et al., 2023) | 85.93 | 93.92 | 71.13 | 86.56 | 91.61 | 88.07 | 84.25 | **85.95** |
| MedSAM (Ma et al., 2024) | _88.18_ | 95.07 | 76.73 | 83.21 | _96.50_ | _97.44_ | 89.09 | 79.23 |
| SAM-Med2D (Cheng et al., 2023) | 84.54 | 95.07 | 71.52 | 73.55 | 91.87 | 95.25 | 89.49 | 75.05 |
| SAM2UNet (Xiong et al., 2024) | 88.11 | 94.57 | _79.20_ | _86.80_ | 92.33 | 91.81 | 89.67 | 82.42 |
| SPG-SAM (Ours) | **91.83** | _95.85_ | **85.97** | **87.62** | 95.00 | **97.87** | **95.57** | _84.90_ |

# 4 EXPERIMENTS

## 4.1 EXPERIMENTAL SETUP

**Datasets**  We evaluate our method on two multi-organ segmentation datasets: the BTCV and the PelvicRT datasets (Landman et al., 2015). The evaluation covers 13 organs in BTCV and 7 anatomical targets in PelvicRT, with performance measured using the mean Dice coefficient (more details in the Appendix).

**Implementation Details**  All experiments are conducted using the ViT-B variant of the SAM model. Input images are resampled to 512×512 resolution before being fed into SPG-SAMs, while predicted logits are resampled to their original dimensions to ensure spatial alignment with raw images. Background classes are excluded from model predictions, and output channels are fixed to predefined anatomical categories. The AdamW optimizer is employed with an initial learning rate

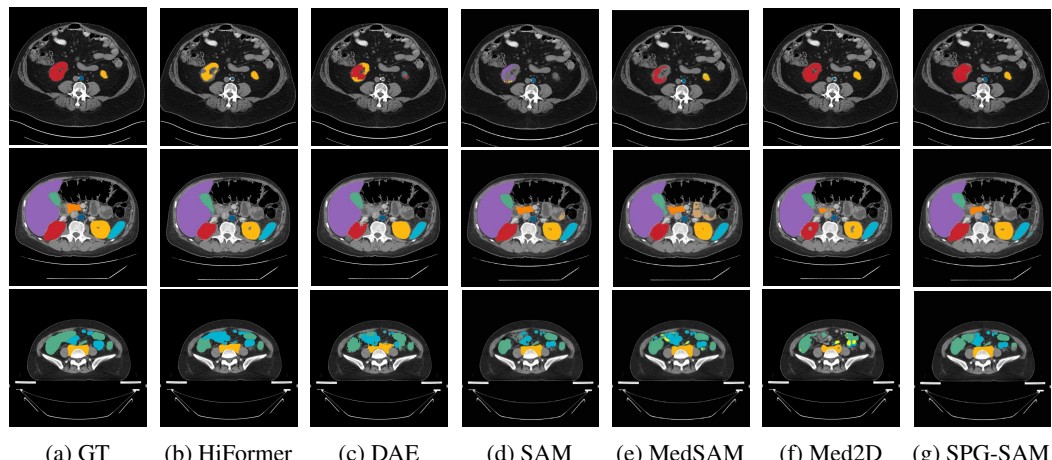

|        |            |         |         |            |           |            |
| (a) GT | (b) HiFormer | (c) DAE | (d) SAM | (e) MedSAM | (f) Med2D | (g) SPG-SAM |

Figure 2: Ground-truth organ segmentation masks and visualization results from several SOTA methods are shown, including HiFormer, DAEFormer (DAE), SAM, MedSAM, SAM-Med2D (Med2D), and our proposed SPG-SAM.

of 0.0001, coupled with a cosine annealing scheduler over 50 training epochs. The loss function incorporated a weighting factor $\alpha = 0.5$, and the rank of LoRA is 8 (more details in the Appendix).

## 4.2 OVERALL PERFORMANCE

**Quantitative Evaluation** We compare SPG-SAM against state-of-the-art (SOTA) methods in multi-class medical image segmentation, including traditional deep learning frameworks (UNet (Ronneberger et al., 2015), TransUnet (Chen et al., 2021), SwinUnet (Cao et al., 2022), MissFormer (Huang et al., 2021), TransDeepLab (Azad et al., 2022), HiFormer (Heidari et al., 2023), DAE-Former (Azad et al., 2023)) and SAM-based extensions (SAM (Kirillov et al., 2023), nnSAM (Li et al., 2023), MedSAM (Ma et al., 2024), SAM-Med2D (Cheng et al., 2023)).

Experimental results on both BTCV and PelvicRT datasets, as shown in Table 1 and Table 2, demonstrate that SPG-SAM exhibits strong competitiveness, setting new SOTA performance. Overall, SAM-based methods surpass traditional approaches, with our method achieving the highest average Dice Similarity Coefficient (DSC) and demonstrating outstanding performance across the majority of medical object categories. While results for "femoral head left" and "small intestine" in the PelvicRT dataset are slightly lower, the overall segmentation accuracy remains highly competitive. For instance, the average DSC improves to 94.27% (BTCV) and 91.83% (PelvicRT), representing a notable increase of 2.1% and 3.65%, respectively, over the second-best performing baselines.

These results indicate that the proposed SPG-SAM can effectively bridge its inherent knowledge gap in medical imaging through rational integration and processing of semantic information. By structurally capturing relationships between medical objects at the whole-image level, SPG-SAM successfully leverages SAM's inherent feature extraction capabilities to enhance prior knowledge learning among medical objects, ultimately achieving performance improvement.

**Qualitative Evaluation** In Figure 2, we present segmentation mask visualizations of several major methods. Compared with other approaches, SPG-SAM demonstrates significant advantages in category sensitivity. For instance, in recognizing symmetric anatomical structures (left kidney & right kidney, left femoral head & right femoral head), traditional segmentation models often confuse isolated symmetrical objects due to insufficient spatial awareness, leading to channel misassignments. SAM-based methods exhibit fewer recognition errors but suffer from increased interference between prompts due to semantic loss, resulting in category over-segmentation (e.g., erroneously segmenting non-existing classes, such as stomach and liver, as well as classes with irregular morphology, e.g., colon & small intestine). In contrast, SPG-SAM leverages both semantic and spatial information to maintain strict sensitivity to existing classes while accurately localizing targets.

Table 3: Ablation study on semantic prompts and graph learning modules.

| Prompt | Method | | | DSC (%) | |
|--------|--------|-----|--------|------|---------|
| | Class Tokens | Graph | Pruning | BTCV | PelvicRT |
| w/o | × | × | × | 76.66 | 79.28 |
| | ✓ | × | × | 79.74 | 80.69 |
| | × | ✓ | × | 80.94 | 80.52 |
| | ✓ | ✓ | × | 80.09 | 79.05 |
| | ✓ | ✓ | ✓ | **91.00** | **86.19** |
| w/ | × | × | × | 87.86 | 87.90 |
| | ✓ | × | × | 92.57 | 88.58 |
| | × | ✓ | × | 87.22 | 86.93 |
| | ✓ | ✓ | × | 92.35 | **91.95** |
| | ✓ | ✓ | ✓ | **94.27** | 91.83 |

Table 4: Impact of the prompt graph learning module insertion position on segmentation performance.

| DSC(%) | $\beta$ | $\alpha$&$\beta$ | $\alpha$ |
|--------|------|--------|------|
| Aorta | **93.76** | 91.16 | 93.30 |
| Gallbladder | 96.12 | 90.07 | **96.42** |
| Kidney (L) | 94.76 | 94.24 | **95.05** |
| Kidney (R) | 92.50 | 93.17 | **94.34** |
| Liver | 93.44 | 89.49 | **94.60** |
| Pancreas | 90.21 | **95.06** | 92.55 |
| Spleen | 95.40 | 93.17 | **96.50** |
| Stomach | **92.44** | 91.85 | 91.36 |
| AVG | 93.58 | 92.28 | **94.27** |

**t-SNE Visualization**  To analyze the effectiveness of the semantic prompt graph learning structure during the inference process, we perform a t-SNE visualization on the mask tokens involved in graph attention, as shown in Figure 4. Here, Figure 4 (a) and (b) represent the visualization of mask tokens in SAM before the cross-attention module and after the MLP, respectively, while Figure 4 (c) and (d) represent the counterparts of SPG-SAM at the same positions. When comparing the figures, it is evident that the distribution of most categories in SAM is relatively uniform without obvious clustering, whereas SPG-SAM exhibits stronger clustering effects and the distinctions between different categories are more pronounced. This highlights the effectiveness of the proposed semantic prompt graph learning in accurately identifying object categories and adaptively facilitating inter- and intra-category feature interactions.

## 4.3 ABLATION STUDY

**Architecture Ablation**  As shown in Table 3, we systematically analyze the impacts of semantic information (class tokens) and graph learning modules on SPG-SAM's medical object segmentation performance under different settings. Experiments demonstrate:

1) Without spatial prompts, when SPG-SAM degenerates into a traditional deep architecture, semantic injection and inter-class graph modeling partially encode anatomical priors (79.28% DSC on PelvicRT). However, fully-connected graph structures introduce interference between present and absent classes, particularly in small-scale data scenarios (PelvicRT), where excessive attention to node relationships over feature learning causes a 0.23% DSC drop (79.28%→79.05%). A category-based graph pruning strategy effectively suppresses noise from absent classes, focusing graph attention on actual anatomical correlations, achieving over 7% DSC improvement across both datasets(80.09%→91.00%, 79.05%→86.19%).

2) With precise spatial prompts, semantic information and spatial constraints form synergistic enhancement: bounding boxes provide strong spatial priors for ROI feature extraction, while graph learning filters anomalies through organ topology constraints (e.g., liver-gallbladder adjacency). Class token integration boosts BTCV DSC by 5.13%(87.22%→92.35%), confirming that semantic context interacts bidirectionally with spatial prompts via self-attention and cross-attention mechanisms to correct semantic-spatial mapping deviations. Removing semantic information disrupts cross-category anatomical correlations (92.35%→87.22%), while disabling sparse graph structures degrades overall performance. This experimentally validates the necessity of tripartite coupling among semantic guidance, spatial constraints, and graph structural reasoning.

**Insertion Position of Graph Attention Network**  To find the optimal insertion point for the prompt graph learning module in the mask decoder, we compare two positions: after the self-attention module ($\alpha$) and after the MLP ($\beta$), designing three configuration schemes (see Table 4 for details).

Experiments on the BTCV dataset demonstrate that single $\alpha$-position insertion achieves the optimal global performance (94.27% Avg DSC), outperforming the $\beta$-position by 0.69%, validating the critical role of deep semantic guidance in multi-organ abdominal segmentation. Dual-position

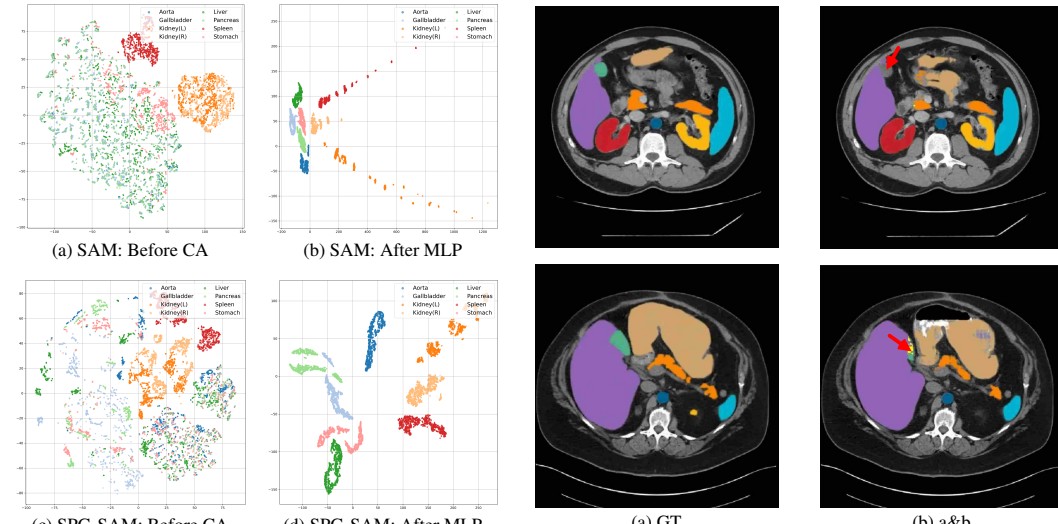

Figure 4: t-SNE visualization of mask token embeddings before and after graph attention. (a) and (c) represent the states before entering cross attention, while (b) and (d) show the outputs after passing through the MLP.

Figure 5: Impact of graph attention module insertion position on segmentation performance. (a) Ground truth; (b) inserted at the dual position. Red arrows mark under-segmentation and mis-segmentation in (b).

insertion leads to a performance decline (92.28%), particularly for the gallbladder (96.42%→90.07%) and liver (94.60%→89.49%), indicating that excessive graph structure attention may impair the features of simple organs through noise interference. As shown in Figure 5, $\alpha\&\beta$ dual-position insertion causes over-segmentation in the gallbladder region, with DSC plummeting from 96.42% ($\alpha$) to 90.07%, suggesting that anatomical structure modeling requires maintaining feature consistency, and complex interventions may weaken feature distinguishability. Comprehensive analysis confirms that introducing semantic prompt map learning at the shallow layer ($\alpha$) enables effective interaction between mask tokens and image embeddings while avoiding inter-class noise from complex graph structures. Therefore, the graph learning component is implemented only at the $\alpha$-position.

Extended ablation results are provided in the Appendix, including experiments and analyses related to the rank size in LoRA, computational efficiency analysis, and qualitative evaluation of interactive prompts.

## 5 CONCLUSION

We propose SPG-SAM, a novel framework that integrates semantic prompt graph learning into the SAM to address the challenges of multi-class medical image segmentation. Unlike existing SAM adaptations that decouple spatial and semantic prompting, SPG-SAM dynamically models inter-class anatomical relationships through a graph attention mechanism while preserving SAM's interactive spatial prompting capabilities. By unifying semantic guidance with spatial constraints and introducing a category-aware graph pruning strategy, our method achieves robust segmentation performance even for ambiguous or co-occurring medical targets. Furthermore, SPG-SAM maintains computational efficiency and clinical practicality by avoiding iterative prompting and redundant computations. It offers a promising direction for leveraging foundation models in medical imaging, balancing automation with precise anatomical reasoning. However, SPG-SAM also has some limitations. For instance, the computational complexity of the graph attention mechanism is relatively high, which may increase training time and resource consumption on large-scale datasets. While SPG-SAM performs well on existing datasets, its generalization capability across different medical imaging modalities still requires further validation. These limitations need to be addressed in future research to further enhance the practicality and adaptability of SPG-SAM.

ETHICS STATEMENT

This work presents SPG-SAM, a framework for multi-class medical image segmentation that integrates semantic prompt graph learning with the Segment Anything Model (SAM). While our research demonstrates significant performance improvements in medical image analysis, we recognize several ethical considerations that warrant careful attention as this technology progresses toward clinical application.

**Human Subjects Research and Data Privacy**  The datasets used in this study (BTCV and PelvicRT) consist of retrospective medical imaging data that were properly anonymized and curated for research purposes. The PelvicRT dataset was collected from XX Hospital with appropriate ethical approvals and patient consent for research use. We implemented strict data protection measures throughout our research, ensuring that no personally identifiable information remains accessible.

**Potential Clinical Applications and Harm Mitigation**  While SPG-SAM shows promising results in medical image segmentation, we emphasize that this technology is intended to assist rather than replace clinical expertise. The framework is designed to enhance physician efficiency by providing accurate segmentation masks, but final diagnostic decisions should remain under human supervision. We acknowledge that erroneous segmentation results could potentially lead to misdiagnosis if used without proper clinical oversight. To mitigate this risk, we recommend implementing confidence metrics and uncertainty quantification in clinical deployments.

**Bias and Fairness Considerations**  Our experiments reveal that SPG-SAM achieves consistent performance across multiple anatomical structures, but we note slight variations in performance for certain organs (e.g., femoral head and small intestine in PelvicRT dataset). These variations may reflect inherent challenges in segmenting particular anatomical structures rather than systematic biases. However, we recognize that the training data may not fully represent the diversity of human anatomy across different populations, age groups, and ethnicities. Future work should include more diverse datasets to ensure equitable performance across all patient demographics.

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

# APPENDIX

## A  MULTI-CLASS MEDICAL IMAGE SEGMENTATION FRAMEWORK VIA SEMANTIC PROMPT GRAPH LEARNING

Figure 6 illustrates the architectural differences between our method and the original SAM in multi-class medical image segmentation. As shown in (1), SAM's interactive architecture inherently operates in a single-target segmentation mode, where each inference only yields a predicted mask for one specific category. This necessitates sequential execution of inferences for all potential medical objects during multi-class segmentation, consuming substantial computational resources and runtime. Meanwhile, the lack of semantic information exacerbates inter-class conflict resolution. Our proposed SPG-SAM addresses these limitations by integrating spatial prompts with semantic prompts during the prompt initialization phase. This mutual mapping mechanism enables precise identification and localization of target medical objects, replacing the original prompt mechanism in subsequent decoding processes. The generated prompt embeddings, combined with image embeddings from the image encoder, are fed into the mask decoder. Here, the semantic prompt graph learning module constructs anatomical relationship graphs to extract anatomically constrained category-specific core features, which actively participate in the final mask generation process. This architecture effectively encodes anatomical prior knowledge while maintaining segmentation efficiency for complex multi-class scenarios.

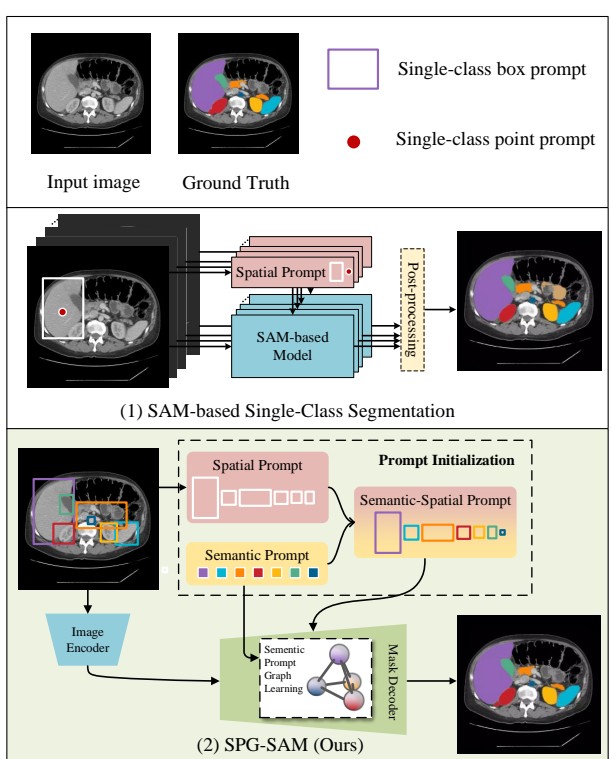

Figure 6: Comparison between SAM-based single-class segmentation and SPG-SAM for multi-class segmentation: (1) SAM-based single-class segmentation: Requires sequential inference for all classes and lacks semantic awareness; (2) SPG-SAM: Achieves multi-class segmentation through single inference while enabling semantics-aware segmentation.

## B  DATASETS

Our method is evaluated on the BTCV multi-organ segmentation dataset and our proprietary PelvicRT dataset. The BTCV dataset originates from the 2015 MICCAI workshop titled Multi-Atlas Labeling Beyond The Cranial Vault (BTCV), and we use its abdominal version. This dataset comprises 2,178 axial 2D abdominal CT images with a train/val/test split ratio of 1,555:338:285. Each image has dimensions of 512×512 pixels, with at least one anatomical category present. The PelvicRT, collected from XX Hospital, contains 474 lower-abdominal CT scan images encompassing 6 organs and 1 clinical target. The dataset is divided with a 9:1:1 ratio for train/val/test split, respectively. All images

are resampled to 1024×1024 pixels, with each slice containing at least one annotated structure, which is verified by at least two senior clinical specialists to ensure label reliability. For evaluation, the mean Dice coefficient is calculated across all 13 BTCV organs (spleen, right kidney, left kidney, gallbladder, esophagus, liver, stomach, aorta, inferior vena cava, portal vein and splenic vein, pancreas, right adrenal gland, left adrenal gland) and 7 PelvicRT anatomical targets (bladder, colon, clinical target volume, left femoral head, right femoral head, rectum, and small intestine).

# C ADDITIONAL EXPERIMENTS

## C.1 THE RANK OF LORA

In model fine-tuning, we employ LoRA to adapt the image encoder while conducting full-parameter fine-tuning on both the prompt encoder and decoder. This approach preserves the original powerful feature extraction capabilities of the image encoder while learning cross-domain medical imaging knowledge, achieving superior fine-tuning results with relatively low computational demands. To explore the optimal rank for the target scenario, we conducted experiments across two datasets. Empirical results demonstrate that rank=8 delivers optimal performance as shown in Table 5. On the BTCV dataset, rank variations exhibit limited impact on segmentation performance (with a narrow range of 0.78%). However, excessively high ranks induced a 2.89% DSC degradation (91.83% → 88.94%) on PelvicRT, which we attribute to introduced noise interference and compromised feature extraction capacity of the native SAM architecture. Conversely, insufficient ranks may lead to inadequate medical feature learning. As visualized in Figure X, different anatomical structures demonstrate heterogeneous rank sensitivity patterns: organs like the colon and spleen benefit from higher ranks, whereas the small intestine and stomach achieve optimal learning at rank=1 with performance degradation at larger ranks. The selected rank=8 represents the balanced configuration that achieves cross-dataset effectiveness.

Table 5: Effect of LoRA rank selection on segmentation performance.

| Datasets | Rank size-DSC (%) | | | |
|---|---|---|---|---|
| | 1 | 4 | 8 | 16 |
| BTCV | 93.95 | 93.49 | **94.27** | 94.21 |
| PelvicRT | 90.69 | 91.26 | **91.83** | 88.94 |

## C.2 COMPUTATIONAL EFFICIENCY ANALYSIS

To comprehensively evaluate model performance, we provide detailed computational efficiency metrics in a comparative analysis (see Table 6). The experiments use SPG-SAM as the baseline model, with two control configurations: one removing the graph learning module (w/o Graph) and another replacing it with a parameter-matched six-layer self-attention module (w/o Graph + 6×Self-Attention). The key findings are as follows:

The three architectures maintain highly consistent computational complexity (approximately 372 GFLOPs), indicating that module adjustments did not significantly affect the overall computational load. Parameter analysis reveals that the baseline model and the self-attention replacement scheme have identical parameter counts (98.7M), while removing the graph module reduces the count to 95.6M, demonstrating that the graph learning module itself contains approximately 3.1M trainable parameters. Inference latency tests show that the graph module introduces only minimal overhead of 4.9ms (baseline 315.5ms vs. 310.6ms without the module), with limited impact on inference efficiency.

Critically, as shown in Table 6, the graph learning module plays an irreplaceable role in enhancing model accuracy: the baseline achieves an average Dice coefficient of 94.27%, removing this module results in a performance drop of 1.7 percentage points to 92.57%, while the parameter-matched self-attention alternative only reaches 92.93%. This confirms that the performance improvement stems not from increased parameter count but from the unique structural advantages of the graph learning module.

Table 6: Computational Efficiency and Performance Comparison of Model Variants. The graph learning module introduces minimal latency overhead (4.9ms) and a small number of parameters (3.1M) but yields a critical and irreplaceable performance gain of +1.7% Dice, significantly outperforming a parameter-matched self-attention alternative.

| Configuration | FLOPs (GFLOPs) | Avg Inference Latency (ms) | Trainable Params | Non-trainable Params | Total Params | Model Size (MB) | AVG DSC (%) |
|---|---|---|---|---|---|---|---|
| Baseline (with Graph) | 372.13 | 315.5 | 9.1 M | 89.7 M | 98.7 M | 394.955 | **94.27** |
| w/o Graph | 372.11 | 310.6 | 5.9 M | 89.7 M | 95.6 M | 382.338 | 92.57 |
| w/o Graph + 6×Self-Attention | 372.27 | - | 9.1 M | 89.7 M | 98.7 M | 394.994 | 92.93 |

## C.3 QUALITATIVE EVALUATION OF INTERACTIVE PROMPTS

To further assess the model's interactive capabilities, we provide a qualitative evaluation of its performance under different prompt types. The experiments primarily utilized bounding box prompts, derived from the minimum enclosing rectangle of the target. Additionally, we conducted experiments with single-point and three-point prompts. The key results are as Table 7.

The results demonstrate that SPG-SAM not only successfully inherits SAM's capability to support multiple prompt types but also exhibits significant performance improvements across all prompting schemes. Specifically: (1) Under the single-point prompt condition, SPG-SAM achieved an average DSC improvement of 4.23% compared to the SAM baseline, with particularly notable enhancements in complex structures such as the stomach (91.71% vs 84.33%) and pancreas (87.99% vs 79.67%); (2) In the three-point prompt scheme, although both models experienced a slight decrease in overall performance, SPG-SAM maintained a clear advantage (90.52% vs 86.46%); (3) Under the optimal bounding box prompt condition, SPG-SAM achieved a top-tier performance of 94.27%, consistently surpassing the baseline method across all anatomical structures. These findings demonstrate the robustness and effectiveness of our method in various interactive scenarios, indicating its ability to fully leverage diverse prompt information to enhance segmentation accuracy.

Table 7: Qualitative Comparison Under Various Interactive Prompts. SPG-SAM consistently outperforms the SAM baseline across all prompt types (single-point/three-point/bounding box) and all anatomical structures.

| Methods | Prompt | AVE DSC (%) | Aorta | Gallbladder | Kidney (L) | Kidney (R) | Liver | Pancreas | Spleen | Stomach |
|---|---|---|---|---|---|---|---|---|---|---|
| SAM | One point | 87.35 | 89.51 | 94.47 | 86.31 | 86.01 | 89.92 | 79.67 | 88.61 | 84.33 |
| | Three points | 86.46 | 88.44 | 91.81 | 84.65 | 85.49 | 89.28 | 82.66 | 87.86 | 81.46 |
| | Bounding Box | 87.86 | 90.06 | 92.45 | 88.56 | 87.76 | 89.25 | 85.47 | 90.12 | 79.20 |
| SPG-SAM | One point | 91.58 | 92.75 | 95.56 | 91.82 | 89.04 | 89.96 | 87.99 | 93.81 | 91.71 |
| | Three points | 90.52 | 91.67 | 94.08 | 92.27 | 92.56 | 92.74 | 86.66 | 92.02 | 82.16 |
| | Bounding Box | 94.27 | 93.30 | 96.42 | 95.05 | 94.34 | 94.60 | 92.55 | 96.50 | 91.36 |

## D STATEMENT ON THE USE OF LARGE LANGUAGE MODELS

During the writing of this paper, the author utilized large language models GPT-4.1 from OpenAI and Deepseek-R1 from Deepseek solely to assist and polish the English writing. Specifically, these models were employed to check and correct grammatical errors, unnatural expressions, and minor punctuation issues in the text originally drafted by the author. Additionally, the models helped to rephrase sentences in order to improve clarity, fluency, and academic tone, while strictly preserving the original technical meanings and scientific content. All core ideas, theoretical frameworks, mathematical derivations, experimental designs, results, analyses, and conclusions are entirely the author's own work. The large language models were used only after the author had completed drafting the core knowledge content, serving purely as writing assistance tools. All suggestions generated by the models were rigorously reviewed, carefully verified, and edited by the author, who assumes full responsibility for the entire content of the published work.

