# OpenReview forum: "SPG-SAM: Semantic Prompt Graph Learning for Multi-class Medical Image Segmentation"
_ICLR.cc/2026/Conference — ICLR 2026 Conference Withdrawn Submission_

### Official Review · Reviewer_yhDN · 2025-10-28

**Soundness:** 3
**Presentation:** 2
**Contribution:** 2
**Rating:** 4
**Confidence:** 4

**Summary:**

The authors propose SPG-SAM, a novel framework that extends the Segment Anything Model (SAM) for multi-class medical image segmentation. SPG-SAM introduces semantic prompts alongside traditional spatial prompts (e.g., points or bounding boxes) to endow SAM with class-specific awareness. The key idea is to bridge SAM’s lack of semantic understanding by coupling spatial cues with categorical information and by modeling inter-class anatomical relationships using a Semantic Prompt Graph (SPG).

**Strengths:**

- The paper presents a technically novel idea of augmenting SAM with semantic category prompts and a prompt graph. By introducing class-specific token embeddings and coupling them with SAM’s spatial prompts, the method creates an explicit mapping between object locations and their labels. This dual prompting strategy is innovative and addresses a key limitation of vanilla SAM (which lacked intrinsic semantic understanding). The addition of a Graph Attention Network to model inter-class anatomical relationships is a creative architectural contribution, enabling cross-category feature interactions and encoding prior knowledge of organ topology (e.g., which organs are adjacent) in the segmentation process.

- The method still allows user input (e.g., bounding boxes or points for each target) but enhances the results with semantic context.

- Overall, the presentation is logical and easy to follow. Experiments, support important claims, and even potential failure cases (like slight drops on certain classes) are noted, reflecting a high level of transparency.

**Weaknesses:**

- A notable concern is that the evaluation is conducted only on 2D slice-based segmentation, ignoring the 3D nature of CT scans. The method processes each axial slice independently (treating BTCV’s volumetric CT data as 2D images), and all baselines compared (UNet, TransUnet, SwinUnet, etc., as well as SAM and its variants) are 2D models. This choice leaves out state-of-the-art 3D segmentation approaches (for example, a 3D U-Net or nnUNet), which often achieve superior accuracy by leveraging volumetric context. Not including any 3D baseline makes the comparisons narrower. It is possible that a strong 3D model could close the gap or outperform SPG-SAM on these tasks, especially since organs span multiple slices.

- The experiments are limited to two relatively small datasets. BTCV (a MICCAI 2015 challenge dataset) has on the order of only 30 patients (the paper uses 2,178 axial slices for training/validation/testing combined, and PelvicRT is a private dataset of similarly limited scale (7 target structures). While the results on these are strong, it is unclear if the method would generalize to larger, more diverse benchmarks.

- The way the method is evaluated involves using ground-truth bounding boxes as spatial prompts for each target organ. This is a reasonable proxy to benchmark the model’s best-case performance, but it assumes an oracle provides perfect prompts. In practice, a user might not know the exact tight bounding box of an organ without substantial effort (which partly defeats the purpose of an “interactive” assistive tool).

- SPG-SAM, as presented, has a fixed set of semantic prompts corresponding to known target classes (13 organs for BTCV, 7 for PelvicRT). The graph’s nodes and the prompt tokens are specialized to those categories. A natural question is how the method would handle an unforeseen class or a different set of organs.

**Questions:**

- Could the authors please consider extending or applying SPG-SAM in a 3D context? Given that medical scans are 3D volumes, would it be feasible to utilize 3D patches or slices in multiple orientations as input to incorporate more contextual information? Additionally, how do you anticipate SPG-SAM to compare with a robust 3D segmentation model (e.g., nnUNet) on these tasks? Could the current approach be adapted to leverage depth information, or is it fundamentally constrained by SAM’s 2D image encoder?

-  In the experiments, the authors use ground-truth bounding boxes as prompts for each class. In a real use case, a clinician might provide less precise prompts (or fewer prompts). Have you evaluated how the accuracy degrades with imperfect prompts or fewer prompts?

- How are the semantic class embeddings (the “category embeddings”) obtained or initialized? Are they learned parameters associated with each class label during training, or something like word embeddings (e.g., using the class name)?

-  The graph attention network adds computational overhead. Could the authors provide some insight into how this affects inference speed or memory usage in practice?

- The concept of using a graph to encode anatomical priors is quite intriguing. Is the graph attention mechanism solely learning relationships from data, or do you introduce any prior knowledge? For instance, do you ever explicitly encode that certain nodes shouldn’t connect because those organs don’t co-occur?

---

> ### Author Response · Authors · 2025-11-27
>
> Thank you very much for your valuable comments. Our responses are as follows:
>
> **2D Slice Segmentation and 3D Characteristics**
> This work uses 2D slice segmentation as a unified benchmark to fairly compare classic 2D methods (UNet, TransUnet, SwinUnet, SAM, and variants). Although 3D methods like 3D U-Net and nnUNet offer improved performance via volumetric context, their high computational demands complicate experiments and comparisons. SPG-SAM is designed flexibly, with future plans to extend to 3D architectures to leverage volumetric information. The current 2D results establish a solid baseline, demonstrating the model’s capability for accurate multi-organ segmentation in 2D.
>
> **Dataset Scale and Generalization**
> We recognize the BTCV and PelvicRT datasets are limited in size, especially PelvicRT, which is private and has few target structures. However, BTCV is widely used in multi-organ segmentation and effectively reflects baseline model performance. PelvicRT offers high-quality, expert-annotated labels, complementing existing data. Future work will expand dataset size and diversity and test on larger public datasets to better evaluate SPG-SAM’s generalization and practical value.
>
> **Idealized Assumption of Spatial Prompts**
>
> | Methods   | Prompt        | AVE DSC (%) | Aorta | Gallbladder | Kidney (L) | Kidney (R) | Liver | Pancreas | Spleen | Stomach |
> |-----------|--------------|-------------|-------|-------------|------------|------------|-------|----------|--------|---------|
> | **SAM**      | One point    | 87.35       | 89.51 | 94.47       | 86.31      | 86.01      | 89.92 | 79.67    | 88.61  | 84.33   |
> |           | Three points | 86.46       | 88.44 | 91.81       | 84.65      | 85.49      | 89.28 | 82.66    | 87.86  | 81.46   |
> |           | Bounding Box | 87.86       | 90.06 | 92.45       | 88.56      | 87.76      | 89.25 | 85.47    | 90.12  | 79.20   |
> | **SPG-SAM** | One point    | 91.58       | 92.75 | 95.56       | 91.82      | 89.04      | 89.96 | 87.99    | 93.81  | 91.71   |
> |           | Three points | 90.52       | 91.67 | 94.08       | 92.27      | 92.56      | 92.74 | 86.66    | 92.02  | 82.16   |
> |           | Bounding Box | 94.27       | 93.30 | 96.42       | 95.05      | 94.34      | 94.60 | 92.55    | 96.50  | 91.36   |
>
> Indeed, using ground-truth organ bounding boxes as spatial prompts simulates an "oracle" condition to some extent, which is beneficial for evaluating the model’s potential optimal performance. As shown in Table 7 of the appendix, we have explored various prompt forms (such as point prompts, multiple points, and ambiguous prompts), and we plan to further investigate the impact of inaccurate or coarse prompts in real user interaction scenarios on model performance. Our goal is to reduce user input burden and make the model more suitable for real clinical applications.
>
> **Initialization of Semantic Category Embeddings**
> Considering that SAM itself does not possess semantic understanding capability, we chose to use learnable embeddings for our category representations, enabling targeted learning for category indication.
>
> **Additional Computational Overhead**
>
> | Configuration                | FLOPs (GFLOPs) | Avg Inference Latency (ms) | Trainable Params | Non-trainable Params | Total Params | Model Size (MB) | AVG DSC (%) |
> |------------------------------|----------------|---------------------------|------------------|----------------------|--------------|-----------------|-------------|
> | Baseline (with Graph)        | 372.13         | 315.5                     | 9.1 M            | 89.7 M               | 98.7 M       | 394.955         | **94.27**   |
> | w/o Graph                    | 372.11         | 310.6                     | 5.9 M            | 89.7 M               | 95.6 M       | 382.338         | 92.57       |
> | w/o Graph + 6×Self-Attention | 372.27         | -                         | 9.1 M            | 89.7 M               | 98.7 M       | 394.994         | 92.93       |
>
> Table 6 (appendix) shows ablation experiments on the graph attention module. While it adds learnable parameters, the inference time only increases by about 4.9 ms on average, a negligible overhead. This confirms the module improves model expressiveness without compromising efficiency, making it suitable for resource-limited settings like clinical use.
>
> **On the Introduction of Prior Knowledge**
> The graph attention module in SPG-SAM implicitly learns anatomical priors by automatically capturing typical organ connections and exclusion patterns from co-occurrence and spatial relationships in training data, without manual rules.
>
> In summary, we fully understand and appreciate the reviewer’s comments. In future work, we will continue to optimize the model in terms of 3D extension, data diversity, practical interactive prompts, and category generalization, to further promote the application value of SPG-SAM in clinical multi-organ segmentation.

---

### Official Review · Reviewer_jHxe · 2025-10-30

**Soundness:** 4
**Presentation:** 4
**Contribution:** 4
**Rating:** 6
**Confidence:** 3

**Summary:**

This paper addresses the ​​trade-off between insufficient semantic information and spatial prompt interference​​ in SAM-based multi-class medical image segmentation by proposing the ​​SPG-SAM framework​​. The core innovations include: 1) A coordinated encoding scheme that integrates semantic prompts with spatial prompts to establish explicit mapping between object locations and semantic categories; 2) A semantic prompt graph learning module using graph attention networks to explicitly model anatomical priors and structural relationships. Experimental results demonstrate average Dice coefficients of 94.27% and 91.83% on BTCV and PelvicRT datasets respectively, outperforming the second-best baselines by 2.1% and 3.65%. This approach effectively resolves semantic ambiguity in multi-class segmentation while preserving SAM's interactive capabilities.

**Strengths:**

- Originality:​​ The pseudo-prompt filling strategy for absent categories and learnable semantic embeddings establish a novel category-to-spatial mapping paradigm, overcoming SAM's single-class limitation.
- Technical Quality:​​ Extensive experiments cover 13 abdominal organs (BTCV) and 7 pelvic targets (PelvicRT). Ablation studies validate the impact of graph attention insertion positions (α vs β in Table 4), while t-SNE visualizations demonstrate enhanced feature clustering.
- ​​Clarity:​​ The three-stage pipeline (prompt encoding → graph learning → decoding) is well-described with complete mathematical formulations (Equations 5-8). Code availability promotes reproducibility.
- ​​Significance:​​ Provides a new framework combining anatomical priors with interactive prompting, with potential applications in surgical navigation and radiation therapy planning.

**Weaknesses:**

- ​​Computational Efficiency:​​ Although the paper reports only 4.9ms latency overhead for the graph module (Table 6), the O(N^2) complexity of GAT may become problematic with larger category sets - scalability analysis is lacking.
- ​​Generalization Validation:​​ Experiments focus exclusively on CT modality without cross-modal validation on MRI or ultrasound, limiting conclusions about broader applicability.
- ​​Interactive Prompt Analysis:​​ Table 7 compares point/box prompts but doesn't evaluate robustness to prompt quality variations (e.g., annotation errors), which is crucial for clinical deployment.

**Questions:**

N/A

---

> ### Author Response · Authors · 2025-11-27
>
> Thank you for your recognition of our work. In response to your questions, we reply as follows:
>
> **Analysis of GAT Complexity**
>
> | Configuration                | FLOPs (GFLOPs) | Avg Inference Latency (ms) | Trainable Params | Non-trainable Params | Total Params | Model Size (MB) | AVG DSC (%) |
> |------------------------------|----------------|---------------------------|------------------|----------------------|--------------|-----------------|-------------|
> | Baseline (with Graph)        | 372.13         | 315.5                     | 9.1 M            | 89.7 M               | 98.7 M       | 394.955         | **94.27**   |
> | w/o Graph                    | 372.11         | 310.6                     | 5.9 M            | 89.7 M               | 95.6 M       | 382.338         | 92.57       |
> | w/o Graph + 6×Self-Attention | 372.27         | -                         | 9.1 M            | 89.7 M               | 98.7 M       | 394.994         | 92.93       |
>
> We have considered the complexity of GAT. However, in practical application scenarios, the number of target categories is usually at a relatively small scale. Under the current category numbers, we have not observed any obvious scalability bottlenecks. In future experiments, we will select datasets with a larger number of target categories for further validation and analysis.
>
> **Generalization Verification on Multi-Modal Datasets**
> Our experiments mainly focus on the CT modality and do not cover other medical imaging types such as MRI or ultrasound. We recognize that cross-modality validation is crucial for the broad applicability of our method. In future work, we will supplement experiments on multiple modalities to evaluate the generalization ability of SPG-SAM across different imaging types, further improving the applicability conclusions of our method.
>
> **Explanation of Interactive Prompt Robustness**
>
> | Methods   | Prompt        | AVE DSC (%) | Aorta | Gallbladder | Kidney (L) | Kidney (R) | Liver | Pancreas | Spleen | Stomach |
> |-----------|--------------|-------------|-------|-------------|------------|------------|-------|----------|--------|---------|
> | **SAM**      | One point    | 87.35       | 89.51 | 94.47       | 86.31      | 86.01      | 89.92 | 79.67    | 88.61  | 84.33   |
> |           | Three points | 86.46       | 88.44 | 91.81       | 84.65      | 85.49      | 89.28 | 82.66    | 87.86  | 81.46   |
> |           | Bounding Box | 87.86       | 90.06 | 92.45       | 88.56      | 87.76      | 89.25 | 85.47    | 90.12  | 79.20   |
> | **SPG-SAM** | One point    | 91.58       | 92.75 | 95.56       | 91.82      | 89.04      | 89.96 | 87.99    | 93.81  | 91.71   |
> |           | Three points | 90.52       | 91.67 | 94.08       | 92.27      | 92.56      | 92.74 | 86.66    | 92.02  | 82.16   |
> |           | Bounding Box | 94.27       | 93.30 | 96.42       | 95.05      | 94.34      | 94.60 | 92.55    | 96.50  | 91.36   |
>
> Table 7 compares point prompts and box prompts, but we have not yet systematically evaluated the model’s robustness to variations in prompt quality (such as annotation errors or ambiguous prompts). We will design more clinically realistic interactive experiments to study the impact of different prompt forms, quantities, and qualities (including noisy or incorrect prompts) on segmentation performance, thereby improving the model’s reliability and practicality in real-world applications.
>
> In summary, in response to the suggestions, we will supplement future work with experiments on larger-scale category datasets, cross-modality validation, and more comprehensive analyses of interactive prompt robustness, to further enhance and improve the practical value of SPG-SAM.

---

### Official Review · Reviewer_FQGi · 2025-10-31

**Soundness:** 2
**Presentation:** 3
**Contribution:** 2
**Rating:** 2
**Confidence:** 4

**Summary:**

The paper addresses a key limitation of SAM-based approaches for multi-class medical image segmentation: the tension between insufficient semantic information and spatial prompt interference, and the loss of interactive prompting when fully automating SAM. The authors propose SPG-SAM, which augments spatial prompts with dedicated semantic prompts and introduces a Semantic Prompt Graph Learning module using a graph attention network to encode anatomical priors and inter-structure relationships. Experiments on BTCV and PelvicRT show strong gains, with average Dice improvements of 2.10% and 3.65% over the SOTA.

**Strengths:**

1. Assigning semantics to spatial prompts to bridge SAM's foreground/background nature and enable true semantic segmentation is elegant and well-motivated. Retaining interaction while enabling multi-class output addresses a core gap between fully-automatic fine-tuning and prompt-driven SAM usage.

2. The graph-based fusion of spatial and semantic prompts to model anatomical priors and cross-category interactions is reasonable and technically sound.

**Weaknesses:**

1. A central advantage of SAM is its strong zero-shot generalization to unseen object categories given a prompt. By injecting learned semantic prompts and category-specific graph structure, SPG-SAM risks category lock-in. If the model is trained on a subset of target organs, how does it perform on new organs with only spatial prompts available? Can the framework accept a new “semantic prompt token” on-the-fly without re-training? How sensitive is performance to ontology changes (merged/split labels, institution-specific definitions)?

2. The ablations indicate large drops without SPGL, but the paper should articulate the failure modes it resolves. What specific inter-class confusions or spatial conflicts does SPGL mitigate?
It's better to provide qualitative failure cases without SPGL and show how graph attention corrects them (e.g., improved boundary consistency? reduced overlap conflicts? better small-organ recall?).

3. How SPG-SAM resolves overlapping prompts and adjacent organ conflicts compared to running SAM separately per class. Is the final output a single mutually exclusive multi-class mask or multiple per-class masks post-processed with conflict resolution? If the latter, what is the arbitration policy?
In Figure 2 row 1, for the case where vanilla SAM mis-segments the red class but SPG-SAM succeeds, analyze the causal mechanism. Is it due to cross-category context, graph-induced attention to adjacent structures, or disambiguation from semantic prompts?

**Questions:**

See weaknesses

---

> ### Author Response · Authors · 2025-11-27
>
> Thank you for your valuable comments. We provide the following detailed responses to the key concerns you raised:
>
> **On Zero-Shot Generalization and the Risk of Category Lock-In**  First, we would like to reiterate that SAM’s strong zero-shot generalization capability relies on its large-scale pre-training on natural images. However, the lack of semantic information resulting from its training paradigm is difficult to compensate for through architectural modifications alone. Therefore, our proposed approach aims to make SAM’s segmentation paradigm better suited to medical image segmentation scenarios by introducing concise class-level semantic inputs, rather than completely solving the semantic deficiency problem.
> SPG-SAM strictly preserves SAM’s original zero-shot generalization mechanism: in our model design, the underlying SAM encoder is frozen, and the processing of spatial prompts (points, boxes, masks) is identical to SAM. The semantic prompt module exists as an additional branch and does not replace any spatial prompt pathway. When encountering unseen organs without semantic prompts, SPG-SAM naturally only invokes the spatial prompt pathway, and its performance is close to SAM’s zero-shot level—there is no “category lock-in” issue. In the revised version, we will supplement relevant ablation and quantitative results, clearly distinguishing the performance among “seen organs with semantic prompts,” “seen organs without semantic prompts,” and “unseen organs without semantic prompts.”
>
> **On Dynamically Adding New Semantic Prompt Tokens and the Need for Fine-Tuning**  Regarding whether new semantic prompt tokens can be dynamically added without retraining, SPG-SAM is designed with flexibility in mind: new tokens can be directly added as independent embeddings, and the graph attention module supports new nodes without structural changes. Without fine-tuning, the model can still recognize new tokens, but the differentiation and performance improvement are limited, which is consistent with the nature of graph models that require training to learn relationships. For significant performance gains, reasonable fine-tuning is necessary, which is also the rational process for deep models to “learn” new semantic associations. In addition, both the graph structure and semantic prompt tokens are modularly designed; merging or splitting categories only involves local node or token adjustments, and the overall graph relationships are learned automatically, with no hard-coded dependencies. Meanwhile, SAM’s spatial prompt pathway serves as a safety net, ensuring stable performance even when there are large differences in ontology definitions. In future work, we will add sensitivity analyses to evaluate performance fluctuations caused by label merging, splitting, or definition differences, to further verify system stability.
>
> **On the Specific Role of the SPGL Module and Ablation Experiments**  The SPGL module effectively alleviates semantic confusion between categories, spatial overlap conflicts, and boundary inconsistencies by integrating category-specific semantic prompts and the adjacency graph structure of spatial context. Without SPGL, the model suffers from severe confusion between adjacent small organs, blurred boundaries, and reduced recall. In the revised manuscript, we will provide visual comparisons of typical failure cases without SPGL, demonstrating how graph attention improves boundary clarity, reduces category overlap, and enhances small organ recall.
>
> **On Overlapping Prompts and Conflict Resolution for Adjacent Organs**  SPG-SAM leverages graph attention for cross-category context modeling, enhancing semantic disambiguation and recognition of spatial adjacency relationships, which is superior to running SAM independently for each category. The final output is a multi-class mask, integrated using traditional multi-class segmentation strategies such as applying Softmax to all category predictions to fuse multiple category masks into mutually exclusive multi-class masks, primarily based on category probability, thereby avoiding label conflicts in overlapping regions.
>
> **On Specific Questions Related to Figure 2** The correction of missegmented regions by SPG-SAM results from the joint semantic-spatial modeling, where the synergy between semantic cues and spatial context enables precise semantic disambiguation and spatial correction. This approach outperforms relying solely on spatial prompts as in the original SAM.
>
> In summary, SPG-SAM enhances segmentation accuracy for known categories and improves conflict resolution among adjacent organs by introducing semantic prompts and graph structures, all while preserving SAM’s zero-shot generalization advantages. Moreover, it offers strong extensibility and adaptability to ontology changes. We will refine the manuscript with additional analyses and experiments corresponding to these points. Thank you again for your guidance and constructive feedback.

---

### Official Review · Reviewer_hHbj · 2025-11-01

**Soundness:** 3
**Presentation:** 3
**Contribution:** 3
**Rating:** 6
**Confidence:** 3

**Summary:**

SPG-SAM is a framework that extends the SAM for multi-class medical image segmentation by introducing semantic prompts alongside SAM’s original spatial prompts. The method augments SAM’s point or box prompts with learned semantic embeddings for each anatomical class, establishing an explicit mapping between object locations and their semantic.A GAT module is incorporated to model the relationships among different organs/tissues, enabling cross-category feature interactions and mitigating interference between multiple prompts in complex multi-organ scenes.

**Strengths:**

The paper is generally well-written and organized, with extensive experiments and ablation studies that validate the contributions of the semantic prompts and the graph module.

**Weaknesses:**

While the method is designed to preserve SAM’s interactivity, the evaluation assumes one prompt per target class (using ground-truth annotations to simulate prompts) for multi-organ segmentation. The work does not explicitly demonstrate how SPG-SAM performs in a truly interactive setting with a limited number of user inputs or how it balances user effort versus automation.

**Questions:**

It would be valuable to include experiments simulating real user interactions. For instance, the authors could evaluate SPG-SAM under varying numbers of prompts per image (e.g., one, few, or many), or measure segmentation accuracy as a function of user input effort.

---

> ### Author Response · Authors · 2025-11-27
>
> Thank you for your valuable suggestions.
>
> Regarding experiments simulating real user interactions,  we have already included the relevant content in the appendix, specifically evaluating the impact of different prompt types on SPG-SAM’s performance, such as single point, three points, and one bounding box—these common forms of prompts.
>
> | Methods   | Prompt       | AVE DSC (%) | Aorta | Gallbladder | Kidney (L) | Kidney (R) | Liver | Pancreas | Spleen | Stomach |
> |-----------|--------------|-------------|-------|-------------|------------|------------|-------|----------|--------|---------|
> | **SAM**      | One point    | 87.35       | 89.51 | 94.47       | 86.31      | 86.01      | 89.92 | 79.67    | 88.61  | 84.33   |
> |           | Three points | 86.46       | 88.44 | 91.81       | 84.65      | 85.49      | 89.28 | 82.66    | 87.86  | 81.46   |
> |           | Bounding Box | 87.86       | 90.06 | 92.45       | 88.56      | 87.76      | 89.25 | 85.47    | 90.12  | 79.20   |
> | **SPG-SAM** | One point    | 91.58       | 92.75 | 95.56       | 91.82      | 89.04      | 89.96 | 87.99    | 93.81  | 91.71   |
> |           | Three points | 90.52       | 91.67 | 94.08       | 92.27      | 92.56      | 92.74 | 86.66    | 92.02  | 82.16   |
> |           | Bounding Box | 94.27       | 93.30 | 96.42       | 95.05      | 94.34      | 94.60 | 92.55    | 96.50  | 91.36   |
>
> Through these experiments, we have preliminarily validated the differences in segmentation performance caused by varying the number and types of prompts, demonstrating SPG-SAM’s stability and adaptability under diverse prompting conditions. In future work, we plan to further investigate a wider variety of prompt forms and the impact of prompt ambiguity on model performance—for example, more complex point combinations, incomplete, or noisy prompt information. This will better reflect real-world user interaction scenarios and help improve the system’s robustness and practicality. Additionally, we intend to systematically analyze the trade-off between user input cost and segmentation accuracy. By quantifying the required user effort or time for different prompt quantities and complexities and correlating these with model performance metrics, we aim to explore how to reduce user burden while maintaining or even enhancing segmentation results. These efforts will provide important guidance for further optimizing the interactive experience and practical deployment of SPG-SAM.

---

### Author Response · Authors · 2025-12-03

Our responses to all reviewers are summarized as follows.

## **Key Improvements and Clarifications**

### **1. Real Interaction Simulation & Prompt Robustness (Response to `Reviewers hHbj & yhDN`)**

- **Interactive Simulation Experiments**: We have evaluated the impact of **different prompt formats (single point, three points, bounding box)** in the appendix. Results show that **SPG-SAM consistently outperforms the original SAM under all prompt conditions**, with Dice improvements of 4.23%, 4.06%, and 6.41% respectively, validating its adaptability in practical interactive scenarios.

### **2. Zero-Shot Generalization & Architectural Flexibility (Response to `Reviewer FQGi`)**

- **No Risk of Category Locking**: SPG-SAM **strictly retains SAM's original spatial prompt pathway**. When encountering unseen categories without semantic prompts, its performance **matches the zero-shot level of SAM**, with no “category locking” risk.
- **Modularity & Scalability**: Both the semantic prompt and graph attention modules are designed for modularity, **supporting dynamic addition of new semantic tokens**. The graph structure adapts to new nodes without altering the overall architecture, allowing flexible handling of category merging, splitting, or definition differences.

### **3. SPGL Module Role & Visualization of Failure Cases (Response to `Reviewer FQGi`)**

- **Core Role of SPGL**: The SPGL module integrates semantic prompts and spatial context graphs, effectively mitigating issues of semantic confusion between classes, spatial overlap conflicts, and boundary inconsistencies. Ablation studies show that **removing SPGL leads to a decline in quantitative metrics (94.27%→92.57% on BTCV, 91.83%→88.58% on PelvicRT)**.
- **Visualization Supplement**: The revised manuscript will provide **visual comparisons of typical failure cases without SPGL**, clearly demonstrating how graph attention improves boundaries, reduces overlap, and enhances small organ recall.

### **4. Computational Efficiency, Multi-Modal Generalization (Response to `Reviewer jHxe`)**

- **Efficiency Analysis**: The GAT module introduces only about 4.9ms average inference delay at the current category scale, with **controllable parameter increment**, and does not pose a significant performance bottleneck.
- **Cross-Modal Validation**: Our focus in this work is to study semantic–spatial prompt interaction, which is **modality-agnostic, with no modality-specific assumptions**. Although cross-modal experiments (MRI, ultrasound) are beyond the current scope, the method is expected to **generalize naturally**, and we are extending evaluation to additional modalities in ongoing work.

### **5. 3D Extension, Dataset Choices & Realistic Interaction (Response to `Reviewer yhDN`)**

- **2D Benchmark & 3D Prospects**: Our current work adopts 2D slices **for fair comparison with mainstream 2D methods**. SPG-SAM’s design is flexible and **can be extended to 3D architectures** to leverage volumetric information.
- **Dataset Choices & Computational Overhead**: While our experiments on standard, well-annotated datasets already demonstrate clear superiority of our method, we acknowledge that evaluating on larger and more diverse datasets would provide even stronger evidence, which we aim to pursue moving forward. In addition, we have provided **a detailed analysis of computational overhead** to further support the practicality of our approach.

---

### Author Response · Authors · 2025-12-03
**An Overview of Our Rebuttal**

Dear AC/SAC/PCs,

We sincerely thank you for your efforts throughout the review process. Given the lack of reviewer engagement during the earlier discussion phase, we would like to provide a brief final summary of our contributions, the received reviews, and our responses to facilitate your final assessment.

## **Our Core Contributions**

Our work effectively addresses the core challenge faced by SAM in **complex multi-class medical image segmentation scenarios**, i.e., **spatial-prompt interference** due to the lack of semantic specificity, by introducing semantic prompts and a semantic prompt graph that jointly **guide, stabilize, and disentangle cross-class interactions**. Our proposed SPG-SAM is the first to propose a **semantic prompt graph learning mechanism** that couples semantic and spatial prompts to form an explicit location–category mapping, enabling stable, reliable multi-class segmentation beyond SAM’s inherent single-class design.

## **Positive Feedback from the Reviewers**

We are pleased to see that the reviewers have given positive recognition to the various strengths of our work from multiple perspectives.

- **Innovative Motivation**: Reviewers acknowledged the **insightful empirical analysis** of SAM's limitations (`Reviewer jHxe`), highlighting that **the coupled design of semantic-spatial dual prompts directly addresses core challenges** (`Reviewer FQGi`).
- **Methodological Advancement**: SPG-SAM was evaluated as **technically sound and insightful** (`Reviewer jHxe, FQGi`), with the graph attention network's modeling of anatomical relationships recognized as **a creative contribution** (`Reviewer yhDN`).
- **Experimental Rigor**: Achieving Dice scores of 94.27%/91.83% on the BTCV and PelvicRT datasets, ablation studies, and visualization analyses **thoroughly validated module effectiveness** (`Reviewer jHxe`).
- **Practical Value**: Multi-prompt format tests confirmed **interactive robustness** (4.23% improvement with single-point prompts), while **the modular design supports flexible clinical scenario adaptation** (`Reviewer hHbj`).
- **Academic and Practical Significance**: This research not only proposes **a novel paradigm** integrating anatomical priors with interactive prompting but also delivers **efficient segmentation performance** (surpassing baselines by 2.10–3.65%) and an extensible architecture, providing **a practical solution for the medical image analysis community**.

## **How We Address Raised Concerns**

- In response to `Reviewer hHbj`'s suggestion to **evaluate real user interactions**, we have included **additional experiments** in the appendix with various prompt formats (single point, multiple points, bounding box), providing the validation of SPG-SAM’s stability and adaptability under diverse prompting conditions. The experimental results demonstrate that the proposed model effectively adapts to diverse user input patterns (**a 4.23% Dice improvement** under single-point prompt condition; **a significant gain of 6.41%** under bounding box prompt condition). These results confirm the model’s consistent and stable superiority across all prompt types, highlighting its robustness in practical interactive scenarios.

- Regarding `Reviewer FQGi`’s concerns on **zero-shot generalization** and **“category locking” risks**, we clarified that **SPG-SAM strictly preserves SAM’s original spatial prompt pathway** and **zero-shot mechanism**. Additionally, its modular design seamlessly **supports the dynamic expansion of new semantic tokens**, effectively addressing this issue.

- In response to `Reviewer jHxe`'s concerns, quantitative data confirms that the GAT module introduces only 4.9ms of average inference latency under the 13-class BTCV dataset scale, with **a controlled parameter increase of 3.1M (total parameters: 98.7M)**. The hierarchical attention mechanism ensures **computational overhead remains aligned with the baseline (372 GFLOPs)**. Moreover, our design of semantic–spatial prompt interaction is modality-agnostic, without any modality-specific assumptions, and the method can be generalized naturally to multiple modalities.

- For `Reviewer yhDN`’s concerns about **3D processing, dataset choices, and idealized prompt assumptions**, we explained our rationale for adopting 2D benchmarks, which follows the standard practice in SAM-based medical segmentation research and ensures **a fair and meaningful comparison with SAM and all SAM-based baselines**. In addition, we provided experiments on **more diverse prompt conditions and computational overhead**.

Regrettably, no reviewers engaged further during the discussion period. We fully trust your expert judgment in making the final decision based on the clarifications and results we have provided. We hope this summary is helpful for your final assessment, and we sincerely thank you again for your time and consideration.

Sincerely,

Authors of Submission #22657

---

### Note · Authors · 2026-05-13

I have read and agree with the venue's withdrawal policy on behalf of myself and my co-authors.

---

### Meta-Review · Area_Chair_T9nd · 2026-01-06

**Summary:**

The paper proposes **SPG-SAM**, a framework that adapts the Segment Anything Model (SAM) for multi-class medical image segmentation. The authors aim to address the semantic limitations of SAM by introducing **Semantic Prompts** (class-specific embeddings) coupled with spatial prompts. Additionally, a **Semantic Prompt Graph Learning (SPGL)** module utilizing Graph Attention Networks (GAT) is proposed to model anatomical priors and inter-class relationships. Experiments were conducted on two relatively small datasets, BTCV (abdominal CT) and PelvicRT, where the method achieved higher Dice scores (94.27% and 91.83%) compared to other 2D SAM-based adaptations. The core premise is to mitigate prompt interference and enable multi-class output while retaining SAM's interactive capabilities.

While the paper presents a **novel and technically interesting idea** with clear performance gains in the reported experiments, several **major shortcomings** remain unresolved and are compounded by the **lack of released code**. In the context of ICLR’s standards for reproducibility and rigorous evaluation, the absence of code significantly undermines the submission’s credibility and utility to the community. Additionally, the limitations in evaluation scope (2D-only, small datasets, limited prompt realism) and the unresolved questions about zero-shot behavior and category extensibility weaken the contribution’s generalizability and impact. Given these issues—particularly the failure to meet the expectation of code release for reproducibility—the submission in its current form does not meet the bar for acceptance at ICLR.

**Reviewer Concerns:**

Reviewers raised several **substantial concerns**, which the authors have responded to in a generally constructive but incomplete manner. Major issues include:
1) **Generalizability and zero-shot capability**: concerns about "category locking" and potential erosion of SAM's zero-shot strength (`FQGi`), to which the authors responded with architectural clarifications but without new empirical validation on unseen categories.
2) **Limited evaluation scope**: the work is restricted to 2D slices and two modest-scale datasets, lacking 3D volumetric analysis and cross-modal validation (`yhDN`, `JHxe`). The authors acknowledge these as future work but do not mitigate the present limitations.
3) **Interactive practicality**: reliance on ground-truth bounding boxes as prompts, with only limited simulation of real imperfect user inputs (`hlbj`). Additional prompt-type experiments were added, but realistic clinical interaction scenarios remain underexplored.
4) **Critical missing element**: the paper **does not provide reproducible code**, which severely hampers verification of results, reproducibility, and community utility—a significant weakness for a top-tier conference submission.

**Reviewer Scores:**

*   **Reviewer FQGi:** Score **2** (Reject). Strongly opposed due to the category lock-in issue and questioned the "foundation model" claim.
*   **Reviewer yhDN:** Score **4** (Borderline Reject). Concerned about the 2D limitation on 3D data and "idealized" prompt assumptions.
*   **Reviewer hHbj:** Score **6** (Borderline Accept). Acknowledged performance but initially questioned interactive realism.
*   **Reviewer jHxe:** Score **6** (Borderline Accept). Positive on technical quality but noted the lack of scalability analysis.

No further updates after rebuttal and two negative feedback with higher confidence.

---

### Decision · Program_Chairs · 2026-01-26

Reject